# On-Chip Polarization Reconfigurable Microstrip Patch Antennas Using Semiconductor Distributed Doped Areas (ScDDAs)

**Rozenn Allanic [1,*]**, **Denis Le Berre [1]**, **Cédric Quendo [1]**, **Douglas Silva De Vasconcellos [2]**, **Virginie Grimal [2]**, **Damien Valente [2]** and **Jérôme Billoué [2]**

1   Laboratoire des Sciences et Techniques de l'Information, de la Communication et de la Connaissance (Lab-STICC), Université de Brest, 29238 Brest, France; denis.le-berre@univ-brest.fr (D.L.B.); cedric.quendo@univ-brest.fr (C.Q.)
2   Laboratoire GREMAN, Université de Tours, 37071 Tours, France; douglas.silva@univ-tours.fr (D.S.D.V.); virginie.grimal@univ-tours.fr (V.G.); damien.valente@univ-tours.fr (D.V.); jerome.billoue@univ-tours.fr (J.B.)
*   Correspondence: rozenn.allanic@univ-brest.fr

**Abstract:** This paper presents two polarization reconfigurable patch antennas using semiconductor distributed doped areas (ScDDAs) as active components. One proposed antenna has a switching polarization between two linear ones, while the other one has a polarization able to commute from a linear to a circular one. The antennas are designed on a silicon substrate in order to have the ScDDAs integrated in the substrate, overcoming the needs of classical PIN diodes. Therefore, the proposed co-design method between the antenna and the ScDDAs permits us to optimize the global reconfigurable function, designing both parts in the same process flow. Both demonstrators have a resonant frequency of around 5 GHz. The simulated results fit well with the measured ones.

**Keywords:** antenna; microstrip; patch antenna; polarization; reconfigurable; ScDDAs; switchable

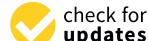



## 1. Introduction

Nowadays, wireless systems are everywhere, and designers have to find new solutions in order to respond to challenges in terms of performance, size and cost. Indeed, the systems have to coexist without creating disturbances for others, all while providing an increase in terms of performance, compactness and a reduction in manufacturing cost. The antennas are one of the main components in these communicating systems and their tunability allows for a reduced size in a multiple standards system. They can be tuned in terms of frequency [1–9] in order to work in several frequency bands of radiation patterns [10,11] in order to modify the orientation of the beam and the polarization [12–21]. Among the various topologies of antennas, microstrip patch antennas are very common antennas used for their compactness, ease of manufacture and low cost.

A novel way to design microwave tunable devices has been developed and consists of a co-design method between active components and passive transmission lines, as in [22,23]. This integration solution with the so-called semiconductor distributed doped areas (ScDDAs) allows us to overcome the needs of classical soldered components. Indeed, the passive devices are designed on a silicon substrate, and thanks to this particular substrate, doped areas can behave electrically via through the substrate thickness, making the device reconfigurable.

Therefore, the passive part is optimized in the same amount of time as a global function. This offers design flexibility, ease of commuting between the working states, a low switching voltage and no parasitic effects between the active and the passive parts. With this monolithic integration of the active elements in a semiconductor substrate, it is also possible to have three working states with a unique DC command [24] and continuous tuning with a triangular-shaped doped area [25].

In this paper, a novel way to co-design on-chip polarization, reconfigurable microstrip patch antennas using ScDDAs are proposed. The flexibility brought by this approach allows for several implementations inducing different kinds of reconfigurability. Hence, the second section presents a switchable patch antenna offering two linear polarizations. A third section proposes a patch antenna with a polarization which commutes from a circular polarization to a linear one.

## 2. Two Linear Polarizations Reconfigurable Antenna

### 2.1. First Antenna Design

The idea is based on a patch antenna design with truncated corners. When the limit conditions are modified in both corners positioned diagonal, this modifies the surface current flow and consequently the polarization of the antenna.

Therefore, the patch antenna is designed on a high-resistivity silicon substrate. This semiconductor substrate allows us to have two semiconductor distributed doped areas (ScDDAs) forming $N^+PP^+$ integrated junctions. These areas act as integrated switches in the substrate thickness depending on their DC bias voltage, modifying the limit conditions onto two corners commuting from open circuits to short circuits. A top view of the antenna design is illustrated in Figure 1a and its side view with the associated technology is shown in Figure 1b. The dimensions of the final design are given in Table 1 with *wa*, which stands for the width of the access line *la*, and *wq* the width of the quarter wavelength line *lq*. *lc* is for the length of the truncated corner, i.e., the smallest dimension of the doped area, while *ldop* is the longest dimension of the doped area and *wdop* is the width of the doped area.

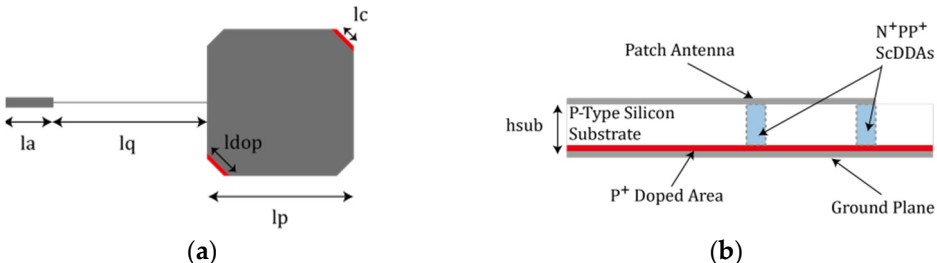

(**a**)                                                                      (**b**)

**Figure 1.** The polarization reconfigurable antenna. (**a**) Top view. (**b**) Side view.

**Table 1.** Dimensions (in mm) of the two linear polarization reconfigurable antenna.

| lp | lq | wq | la | wa | lc | ldop | wdop | hsub |
|-----|-----|-------|-----|------|------|------|------|-------|
| 8.7 | 9.2 | 0.016 | 0.3 | 0.56 | 1.41 | 1.84 | 0.3 | 0.675 |

### 2.2. First Antenna Simulations

The simulations were performed using a full-wave electromagnetic simulator HFSS$^{TM}$ from Ansys. In the OFF state (considering the junctions without bias voltage), the doping areas are neglected and the resistivity of $\rho$ = 2500 $\Omega$.cm is taken into account in the loss tangent calculation thanks to Equation (1) [26].

$$tan\delta = \frac{1}{\rho\omega\varepsilon_0\varepsilon_r} + 0.0018 \tag{1}$$

In the ON state (when a direct bias voltage is applied to the junctions), the junctions are simulated with a homogenous resistivity of $\rho$ = 0.5 $\Omega$.cm in the whole substrate thickness under the surfaces of the doped areas.

Figure 2 presents the simulated reflection coefficient in both states. In the OFF state, the −10 dB bandwidth is between 4.86 GHz and 4.95 GHz, whereas in the ON state, the −10 dB bandwidth is between 4.95 GHz and 5.03 GHz. Figure 3 illustrates the electrical field in both states. In the OFF state, there is no short circuit in the substrate, and the polarization is a linear one, parallel to the slots, so the patch radiates as a classical one. In the ON state,

the ScDDAs are some short circuits in the substrate, creating a perturbation in the current flow and modifying the polarization way to a linear at 45° from the original one.

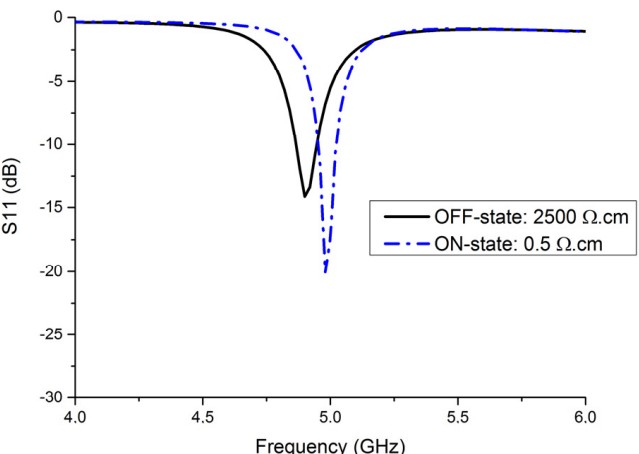

**Figure 2.** The simulated reflection coefficient in both states of the two linear polarization antennae.

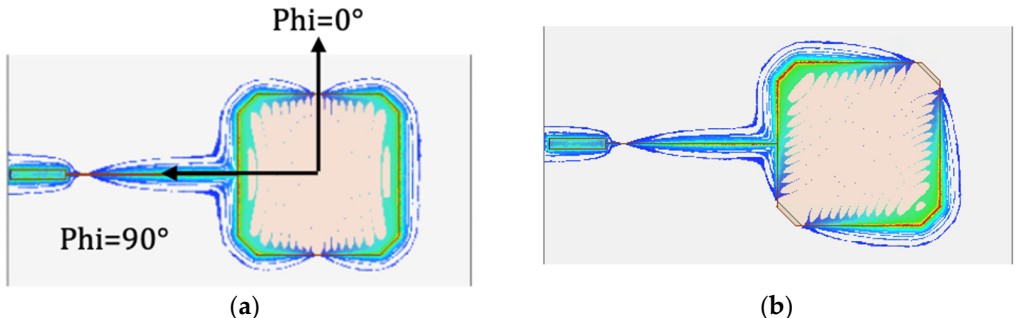

(**a**)

(**b**)

**Figure 3.** A simulated EM field. (**a**) Horizontal Linear Polarization. (**b**) 45° Linear Polarization.

Figure 4 shows the normalized radiation patterns in both states. The diagram is omnidirectional as a classical patch with a difference between the co- and cross-polarization higher than 45 dB in the OFF state and 12 dB in the ON state.

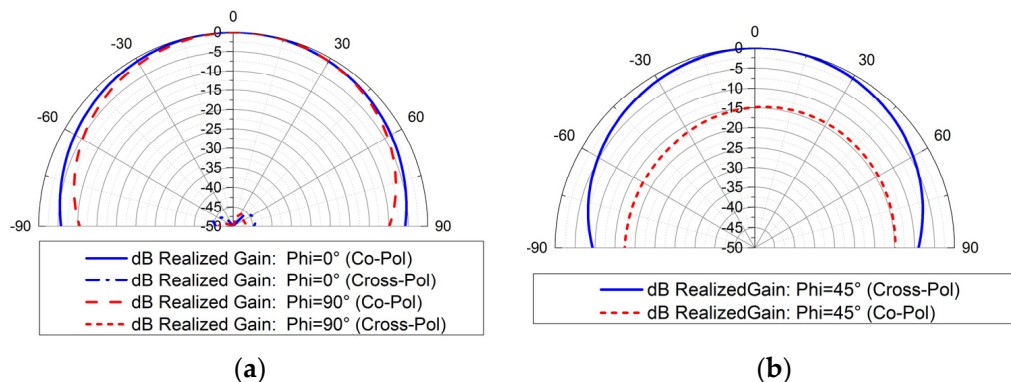

(**a**)

(**b**)

**Figure 4.** Simulated radiation patterns of the realized gain. (**a**) In the OFF state at Phi = 0° and at Phi = 90°; (**b**) in the ON state at Phi = 45°.

### 2.3. First Antenna Fabrication

A high-resistivity silicon P-type substrate was selected for manufacturing to minimize the loss tangent and the cost. The P$^+$ and N$^+$ areas are doped with a solgel solution and diffusion technique to reach around $10^{19}$ atoms/cm$^3$ with Boron and Phosphorus atoms, respectively. The depth of the two junctions is around 3 µm. The process steps are described in [27].

Indeed, only two masks are required for the manufacturing, as illustrated in Figure 5. Figure 6 shows a photograph of the fabricated demonstrator. In fact, the chosen SMA connector allows for measurement of a device with a thickness of 1.2 mm, whereas the silicon substrate thickness used is 0.675 µm, which is why an aluminum plate was put under the antenna to ease the placement of the connector, thus making the measurement possible.

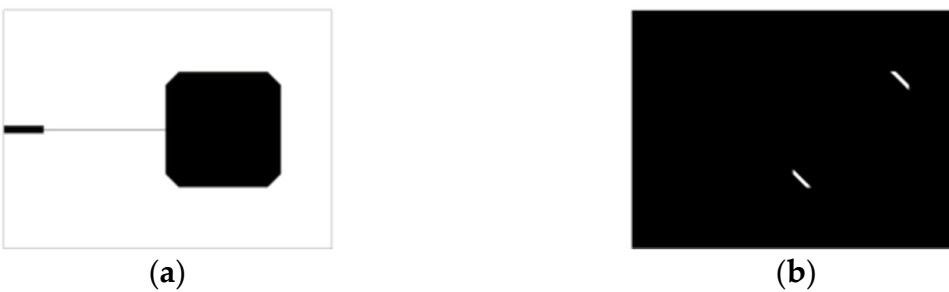

(**a**) (**b**)

**Figure 5.** (**a**) The metallization mask. (**b**) The doping mask.

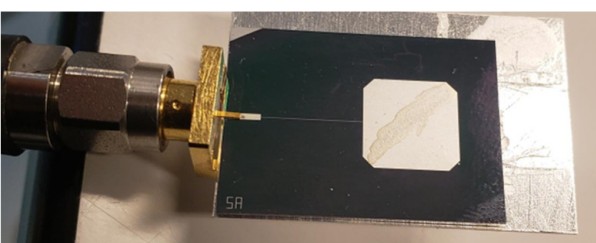

**Figure 6.** A photograph of the first prototype.

### 2.4. First Antenna Measurement

Firstly, the $S_{11}$ parameter was measured using a vector network analyzer (VNA ZVA 67 from Rohde & Schwarz®). The antenna commutation is provided by applying a DC bias voltage with a DC source connected directly to the analyzer. The DC bias voltage is applied with the RF signal, and due to the kind of junctions, a negative voltage is required to apply direct bias to them. A bias voltage of −6 V is applied at the output of the source. Figure 7 presents the measured results of the reflection coefficient in the OFF and ON states. In the OFF state, the resonant frequency is 4.93 GHz with a $S_{11}$ level lower than −22 dB. In the ON state, the resonant frequency is 4.96 GHz with a $S_{11}$ level lower than −10.2 dB.

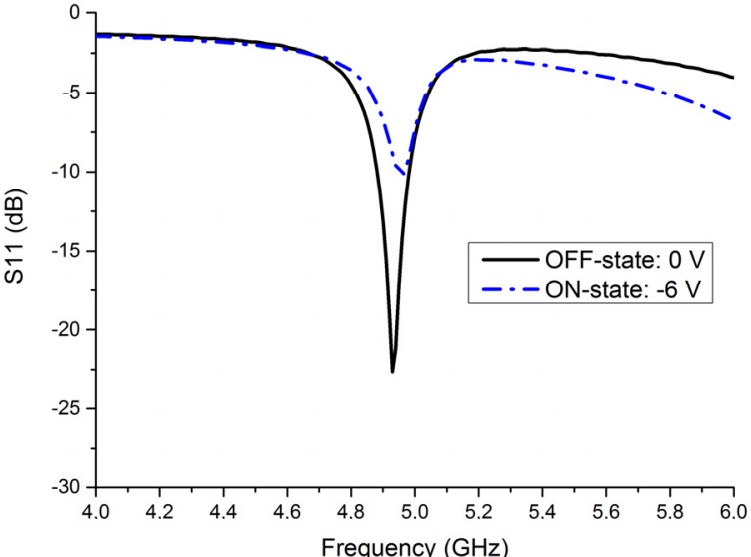

**Figure 7.** Measured results of the two linear polarization reconfigurable antenna in both states.

Secondly, the antenna was measured in an anechoic chamber, as illustrated in Figure 8. Two supports were made in 3D-printing technology; the white one is for the 0° and 90° radiation patterns measurement and the red one is added for the 45° radiation pattern measurement.

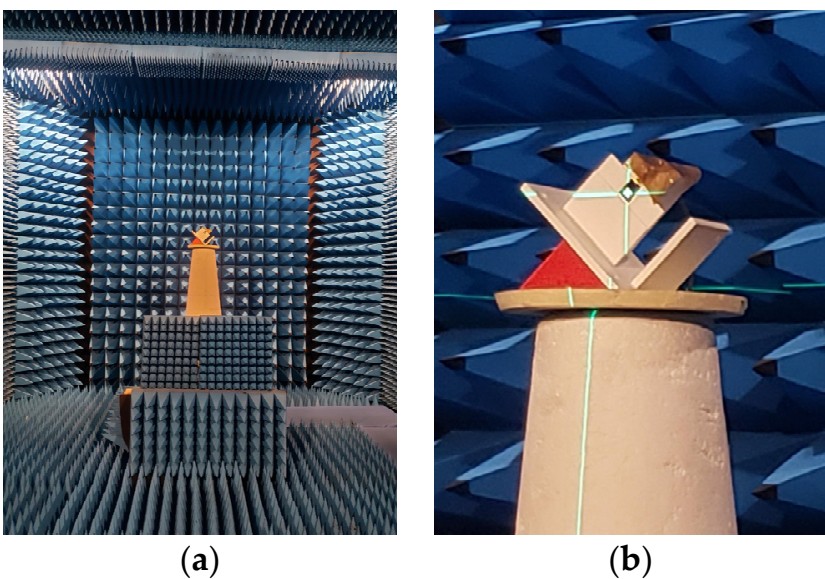

**Figure 8.** (**a**) An antenna under measurement in the anechoic chamber. (**b**) A close-up of the positioner.

Figure 9a shows the co-polarization and the cross-polarization in both 0° and 90° planes for the OFF state. The shapes of the measured radiation patterns are identical to those simulated even if in the OFF state, and the cross-polarization level is higher than the simulated one.

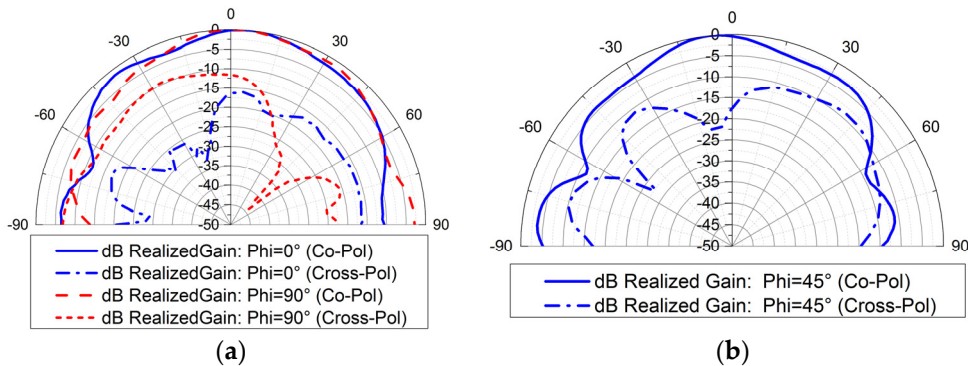

**Figure 9.** (**a**) Measured radiation patterns of the realized gain. (**a**) In the OFF state at Phi = 0° and at Phi = 90°; (**b**) in the ON state at Phi = 45°.

This difference is due to the measurement setup. Indeed, the antenna is not soldered to the connector but only inserted between the pins of the connector and maintained by adding an aluminum plate. This induces misalignments between the emitting antenna and the patch antenna, which explains the higher cross-polarization level. Figure 9b shows the co-polarization and the cross-polarization in the 45° plan for the ON state. The shape and the difference between the co- and cross-polarization are almost the same as in the simulated results.

## 3. Circular to Linear Polarization Reconfigurable Antenna

### 3.1. Second Antenna Design

Figure 10 presents a second demonstrator able to commute from a circular polarization to a linear one. The patch antenna has two truncated corners to assure the circular

polarization in the OFF state, and two doped areas are located in the other two corners to modify the current flows by adding two electrical short circuits in the substrate in the ON state. Table 2 summarizes the antenna and the doped area dimensions with *wa*, which stands for the width of the access *la*, and *wq* the width of the quarter wavelength line *lq*. *lc* is for the length of the truncated corner, *ldop* is for the longest dimension of the doped area and *wdop* is for the width of the doped area.

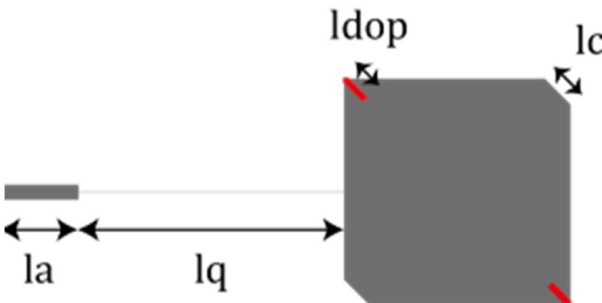

**Figure 10.** A top view of the circular to linear polarization switchable antenna design.

**Table 2.** Dimensions (in mm) of the two circular to linear polarization reconfigurable antennae.

| lp | lq | wq | la | wa | lc | ldop | wdop |
|----|----|----|----|----|----|------|------|
| 8.7 | 10.2 | 0.016 | 0.3 | 0.56 | 1.41 | 1 | 0.2 |

*3.2. Second Antenna Simulations*

This antenna was designed to have a resonant frequency of around 5 GHz in both states. Figure 11 shows the simulated results of the reflection coefficient. In the OFF state, the −10 dB bandwidth is between 4.77 GHz and 4.96 GHz and in the ON state, with a resistivity in the substrate thickness of 0.5 $\Omega$.cm, the −10 dB bandwidth is between 4.9 GHz and 4.98 GHz.

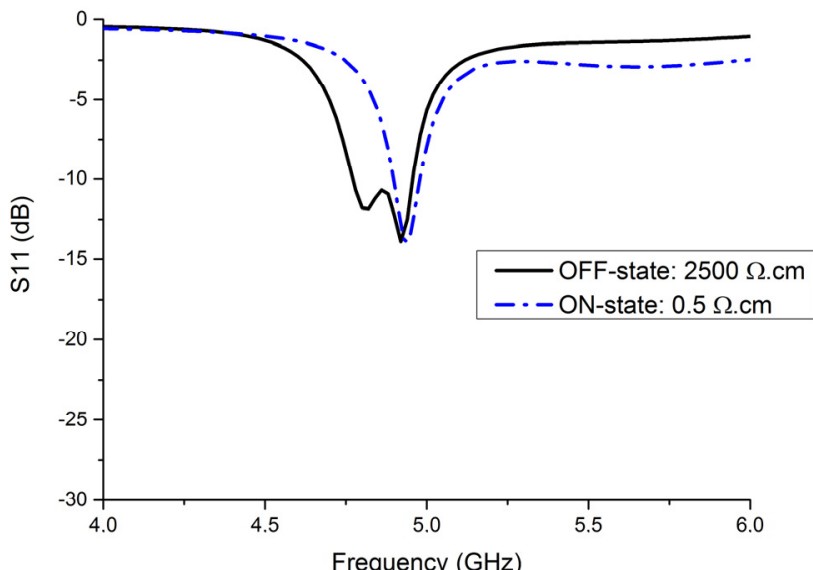

**Figure 11.** $S_{11}$ simulated results of circular to linear polarization reconfigurable antennae in both states.

Figure 12 presents the simulated radiation patterns in both states. In the OFF state, the co- and cross-polarizations of the phi angle have the same level, which shows that the polarization of the antenna is a circular one. The axial ratio is lower than 2.7 dB. In the

ON state, at phi = 45°, the difference between the co- and cross-polarization is higher than 12 dB, which is a linear polarization.

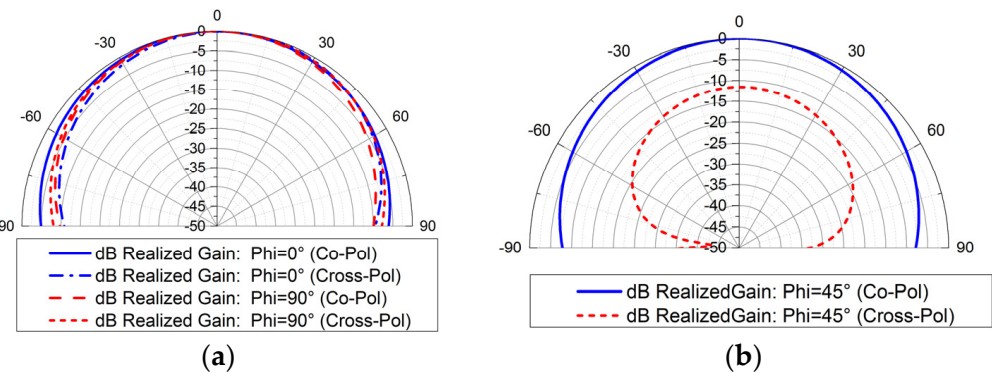

**Figure 12.** Simulated radiation patterns of the realized gain. (**a**) In the OFF state in a circular polarization; (**b**) in the ON state in a linear polarization.

### 3.3. Second Antenna Measurement

A demonstrator was manufactured, and Figure 13 shows the antenna with its SMA connector and its aluminum plate to assure the best maintenance as possible. The measured results are presented in both states in Figure 14. In the OFF state, the -10 dB bandwidth is between 4.75 GHz and 4.86 GHz, with a reflection coefficient of −10 dB at 4.96 GHz. In the ON state, the −10 dB bandwidth is between 4.89 GHz and 4.99 GHz.

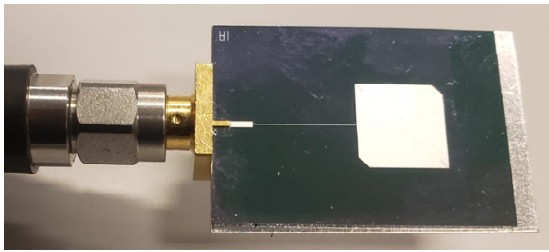

**Figure 13.** A photograph of the second antenna.

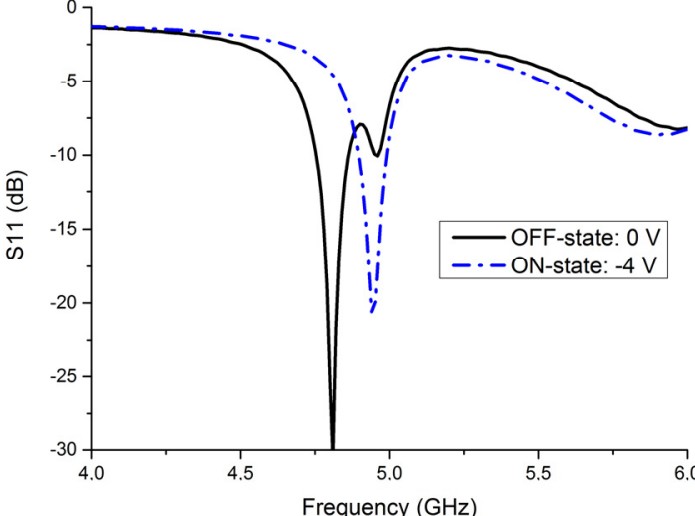

**Figure 14.** Measured results of the circular to linear polarization reconfigurable antenna.

The radiation patterns of this antenna were measured in both states in an anechoic chamber. The measured results are presented in Figure 16a,b in the OFF state with a circular polarization and in the ON state with a linear polarization, respectively.

Therefore, except for a slight rise in the reflection coefficient in the bandwidth in the OFF state, there is a good agreement between the simulated and measured results, as shown in Figure 15.

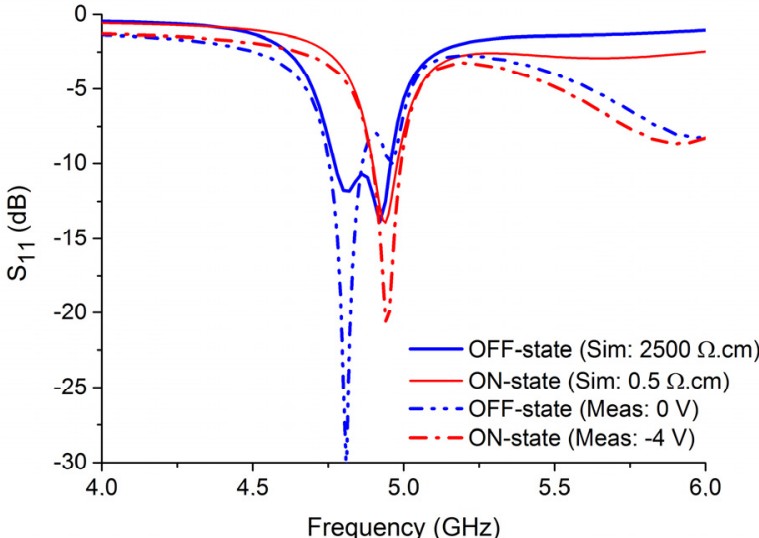

**Figure 15.** A comparison between simulated and measured results in both states.

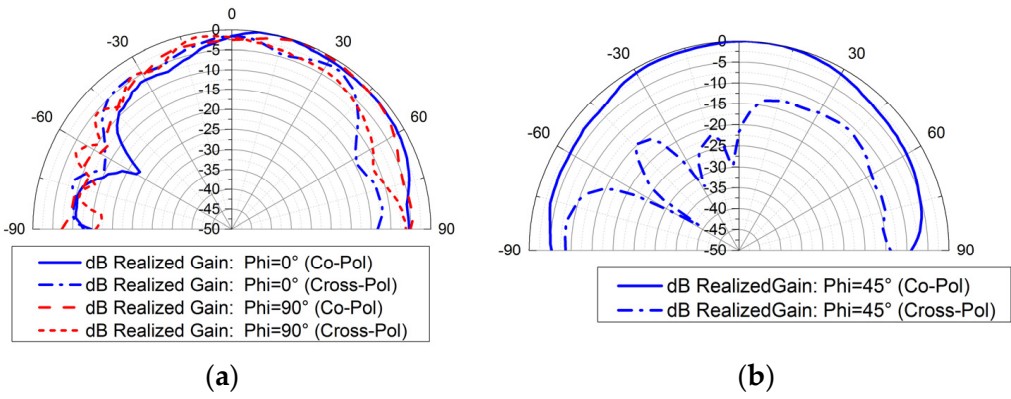

**Figure 16.** Measured results. (**a**) In the OFF state, in a circular polarization; (**b**) in the ON state, in a linear polarization.

## 4. Discussion

Table 3 shows a comparison between previous polarization reconfigurable antennas and this work. The antennas are either circularly polarized (CP) in a right-hand circular polarization (RHCP) and/or left-hand circular polarization (LHCP) configuration or linearly polarized (LP) in a horizontal polarization (HP) and/or in a vertical polarization (VP) and/or in a 45° polarization. Few polarization reconfigurable antennae can be found in the literature above 3 GHz. The reconfigurability is achieved thanks to PIN diodes, causing disturbances that increase with the increase in frequency. The proposed patch antennas are designed in a simple manufacture process with only two photolithography masks, allowing small and accurate dimensions of the microstrip lines and particular shapes of the doped areas, permitting an increase in frequency. Moreover, the codesign method given by the choice of a silicon substrate allows for on-chip reconfigurable antennas with integrated junctions in the substrate without requiring additional components.

**Table 3.** A comparison between previous polarization reconfigurable antennae and this work.

| Ref. | Topology of the Antenna | Technology | Freq. (GHz) | Polarization | Active Components |
|---|---|---|---|---|---|
| [12] | Patch | Multilayers on PCB | 2.4 | RHCP/LHCP/LP | 4 PIN diodes |
| [13] | Patch | PCB | 1.6 | CP/LP | 2 PIN diodes |
| [15] | Circular patch | PCB | 2.4 | RHCP/LHCP/LP | 4 PIN diodes |
| [18] | Etched ring shape slot | Multilayers on PCB | 3.1 | RHCP/LHCP/VP/HP | 4 PIN diodes |
| [19] | Patch | PCB | 2.4 | RHCP/LHCP/LP | 2 PIN diodes |
| [20] | U-slot | PCB | 5.7 | RHCP/LHCP/VP/HP | 2 PIN diodes |
| [28] | Patch | Multilayers on PCB | 2.4 | RHCP/LHCP/LP | 3 PIN diodes |
| This study $A_1$ | Patch | Silicon | 4.9 | LP (90°)/LP (45°) | 2 Integrated ScDDAs |
| This study $A_2$ | Patch | Silicon | 4.9 | CP/LP | 2 Integrated ScDDAs |

## 5. Conclusions

Polarization reconfigurable patch antennas were proposed in this paper as part of a monolithic integration technology. One of the antennae commutes between two different linear polarizations (0° to 45°), while the other one switches from a circular to a linear polarization. The active elements, which are some integrated ScDDAs, overcome the need of classical PIN diodes which limit the parasitic effects. Moreover, the co-design method offers flexibility in the ScDDAs shape (such as the complex shape of the proposed demonstrator), allowing for an optimization of the two commutation states. Two demonstrators were proposed and analyzed to show an overview of the possibilities. The measured results validate the proof of concept. These antennas with polarization reconfigurability could be applied to RADAR applications in a suitable band.

**Author Contributions:** Conceptualization, R.A.; methodology, R.A.; simulation and characterization, R.A.; validation, R.A., D.L.B. and C.Q.; formal analysis, C.Q.; manufacture, D.S.D.V., V.G., D.V. and J.B.; writing—original draft preparation, R.A.; writing—review and editing, D.L.B. and C.Q.; supervision, D.L.B., C.Q. and J.B. All authors have read and agreed to the published version of the manuscript.

**Funding:** This work was partly supported by both CERTeM Technological and RENATECH network French Platforms. This publication is also supported by the European Union through the European Regional Development Fund (ERDF), by the Ministry of Higher Education and Research and by Brest Métropole, Brittany, through the CPER Project SOPHIE/STIC & Ondes.

**Institutional Review Board Statement:** Not applicable.

**Informed Consent Statement:** Not applicable.

**Data Availability Statement:** Not applicable.

**Acknowledgments:** The authors would like to thank the TECHYP platform (the High Performance Computing Cluster of the Lab-STICC) for device simulations. The authors would also like to thank ENSTA-Bretagne engineering school for the use of its anechoic chamber and Fabrice Comblet for his help.

**Conflicts of Interest:** The authors declare no conflict of interest.

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
