# Peer review of "On-Chip Polarization Reconfigurable Microstrip Patch Antennas Using Semiconductor Distributed Doped Areas (ScDDAs)"

_electronics, doi:10.3390/electronics11121905_

Round 1

Reviewer 1 Report

The paper is fairly written, and it will be of interest to radio and wireless communication systems. However, there are some minor comments which needs to be addressed by the authors to add value to the paper. I will mention some of them. 

1. Title caption for section 2.2 and 3.2 are the same, authors are advised to change the caption to distinguish them.

2. Title caption for section 2.1 and 3.1 are the same, authors are advised to change the caption to distinguish them

3.Please justify the statement on page 7 line 157... 'if the coefficient in the bandwidth is in the OFF-state' why?

4. Please provide a statement on the application of the deign in the abstract and conclusion section. 

Author Response

Comments of reviewer 1:

The paper is fairly written, and it will be of interest to radio and wireless communication systems. However, there are some minor comments which needs to be addressed by the authors to add value to the paper. I will mention some of them. 

Answer: thank you very much for your kind comments and for your advises to improve the quality of the paper.

  1. Title caption for section 2.2 and 3.2 are the same, authors are advised to change the caption to distinguish them.

Answer: Thank you for your comment, the captions are now all different.

  1. Title caption for section 2.1 and 3.1 are the same, authors are advised to change the caption to distinguish them

Answer: Thank you for your comment, the titles are now all different.

3.Please justify the statement on page 7 line 157... 'if the coefficient in the bandwidth is in the OFF-state' why?

Answer: Thank you for your comment, in the ON-state there is a good agreement between the simulated and measured results. In the OFF-state, the difference is probably due to the connector placement and the approximation in the doping quantities.

  1. Please provide a statement on the application of the deign in the abstract and conclusion section. 

Answer: Thank you for your valuable comment, this antenna concept could be applied to RADAR applications in a suitable band. We added some details on the application in the conclusion.

Reviewer 2 Report

This work by Allanic et. al. presents a technology that enables polarization reconfigurable patch antennas with semiconductor distributed doped areas (ScDDAs). By integrating the ScDDAs in the substrate, this design can achieve good performance which was realized with classical PIN diodes. The co-design method allows a flexibility in the ScDDAs shape for further optimization of the two commutation states. The experiments are well designed and the model in numerical simulation is reasonable. 

Author Response

Comments of reviewer 2:

This work by Allanic et. al. presents a technology that enables polarization reconfigurable patch antennas with semiconductor distributed doped areas (ScDDAs). By integrating the ScDDAs in the substrate, this design can achieve good performance which was realized with classical PIN diodes. The co-design method allows a flexibility in the ScDDAs shape for further optimization of the two commutation states. The experiments are well designed and the model in numerical simulation is reasonable. 

Answer:

Thank you for your encouraging comments. Some Figures and explanations about the method were added to improve the understanding. Other modifications were done in the conclusion and the results presentation to improve the paper quality.

Reviewer 3 Report

This is an interesting work with integrated configurable antenna on high resistance Si substrate. 

For the key component of ScDDA (0.675 mm through the silicon wafer) , no much detail is provided for the fabrication of the device, such as anneal temperature and time, etc.

Table 3 compared this work with previous work. This is good. However, some performance related parameters are not included. for example, antenna gain, power loss, bandwidth, tunable range, etc.

Figure numbers is not in order, 4,5,6 is after 8 and 9.  no figure number 7, 10, 11 with 12 as the last figure. below are referred in text with figure number more than 12.

- Line 150, refer to fig. 13. 

- Line 158 refer to fig. 15.

- Line 161 refer to fig. 16

Author Response

Comments of reviewer 3:

This is an interesting work with integrated configurable antenna on high resistance Si substrate. 

Answer: Thank you for your encouraging comments.

For the key component of ScDDA (0.675 mm through the silicon wafer) , no much detail is provided for the fabrication of the device, such as anneal temperature and time, etc.

Answer: Thank you for your suggestion. We added the following sentence and the associated reference: The whole process steps are described in [28].

  1. Allanic et al., "A Novel Synthesis for Bandwidth Switchable Bandpass Filters Using Semi-Conductor Distributed Doped Areas," in IEEE Access, vol. 8, pp. 122599-122609, 2020.

Table 3 compared this work with previous work. This is good. However, some performance related parameters are not included. for example, antenna gain, power loss, bandwidth, tunable range, etc.

Answer: Thank you for your valuable comment. In the reference papers the antenna gain, power loss, bandwidth are not always given which lead to the impossibility to include these parameters in the comparison table. However, the kind of reconfigurability is given and included in the table, such as RHCP/LHCP/VP/HP, same as the technology and the active elements.

Figure numbers is not in order, 4,5,6 is after 8 and 9.  no figure number 7, 10, 11 with 12 as the last figure. below are referred in text with figure number more than 12.

- Line 150, refer to fig. 13. 

- Line 158 refer to fig. 15.

- Line 161 refer to fig. 16

Answer: Thank you for your comment. We are quite sorry about this problem. Indeed, there was a problem with figure numbers and figure references. All numbers were corrected.